# Essential Oil Content and Compositional Variability of *Lavandula* Species Cultivated in the Mid Hill Conditions of the Western Himalaya

**DOI:** 10.3390/molecules27113391

**Published:** 2022-05-25

**Authors:** Shalika Rathore, Rakesh Kumar

**Affiliations:** 1Agrotechnology Division, CSIR—Institute of Himalayan Bioresource Technology, Post Box No. 6, Palampur 176 061, India; shalikarathore13@gmail.com; 2Academy of Scientific and Innovative Research (AcSIR), Ghaziabad 201 002, India

**Keywords:** lavender, lavandin, linalyl acetate, linalool

## Abstract

The increase in the utilization of *Lavandula* essential oil in industries led to an impressive rise in the demand for quality essential oils. However, a post-harvest drying of *Lavandula* species can be a decisive factor to determine the quantity and quality of essential oil. The study was conducted in western Himalayan conditions to assess the essential oil content and composition of two *Lavandula* species viz., lavender (*Lavandula angustifolia* Mill.), and lavandin (*Lavandula* × *intermedia* Emeric ex Loisel), at four different drying duration (0 h, 24 h, 48 h and 72 h after the harvest). The higher growth attributes viz., plant height (71.7 cm), ear length (8.8 cm), number of spikes (18.1), and number of flowers per ear (47.5) were higher in lavandin, while the number of branches (17.1) was higher in lavender. Essential oil content (%) and moisture reduction (%) were significantly higher at 72 h than at 0 h. The major components of lavender and lavandin essential oil were linalool (33.6–40.5%), linalyl acetate (10.8–13.6%), lavandulyl acetate (2.8–14.5%), and linalyl propionate (5.3–14.1%) in both the *Lavandula* species. There was a decreasing trend in linalool and an increasing trend in linalyl acetate content in lavandin, with an increase in drying duration up to 72 h; while in lavender, no regular trend was observed in linalool and linalyl acetate content. It was observed that linalool and linalyl acetate levels were the highest at 24 and 0 h of drying in lavender and lavandin, respectively, and essential oil extraction can be done according to the desire of the constituent at varied drying duration.

## 1. Introduction

The usage of medicinal and aromatic plants (MAPs) since prehistoric times have been well known, but these are recently recognized around the world as economical crops for high-valued essential oil [1]. The essential oil-producing plants belong to different genera with families including Lamiaceae, Asteraceae, Apiaceae, Alliaceae, Rutaceae, Poaceae, and Myrtaceae. One of the important plant families is “Lamiaceae”, which has been reported to contain 236 genera and 6900–7200 species. The genus *Lavandula* is of major significance inthe Lamiaceae family, covering more than 40 species and many subspecies. The principal cultivated species for aromatic oils are fine lavender (*Lavandula angustifolia* Mill.), spike lavender (*L. latifolia*), and lavandin (*L.* × *intermedia* Emeric ex Loisel), a sterile hybrid of *L. angustifolia* × *L. latifolia* [2]. The main production country of lavender is Bulgaria, followed by France; the top producer of lavandin essential oil is France, followed by Spain [3]. The essential oil composition of *Lavandula* species almost has a similar chemical profile, but the components are usually present in varied proportions. The presence of a few essential oil components, viz., camphor, linalool, and linalyl acetate is the common criterion for quality determination of essential oil of lavandin [4]. The lavender essential oil contains linalyl acetate (25–45%), linalool (25–38%), and camphor (0.5–1.0%), while lavandin essential oil contains linalyl acetate (28–38%), linalool (24–35%), and camphor (6–8%) according to the ISO 3515:2002 and ISO 8902:2009 standards, respectively. Lavender essential oil is used in pharmaceuticals, perfumery, and cosmetic industries because of its higher linalool and linalyl acetate content, while essential oil of lavandin is usually utilized in industrial and domestic cleaning products, hygiene, and detergents owing to higher camphor levels [2]. Although the yield of essential oil is higher in lavandin (120 kg ha^−1^) compared to lavender (40 kg ha^−1^) [5], according to International market price of 2018, essential oil quality and price of lavender (100.89–178.03 USD kg^−1^) is higher than lavandin (22.6 USD kg^−1^) [2,3,6]. The lavender essential oil is much desired in the marketplace, although the amount of production is low, while lavandin essential oil is different in quality due to its relatively higher camphor content [6]. The commercial and industrial cultivation of lavender and lavandin has been raised rapidly during the last years due to increased world demand. The essential oil market of lavender was 530.5 USD Million in 2020 and is expected to reach 864.7 USD million by 2025 with a CAGR (compound annual growth rate of 10.3% from 2020 to 2025 [7]. *Lavandula* essential oil has some uses such as reliever of stress, anxiety, insomnia, migraines, and depression [8,9,10,11]; it also has biological properties, viz., antimicrobial, antioxidant anticholinesterase [12,13,14]. The qualitative and quantitative composition of essential oil of *Lavandula* species depends on genetic makeup [15], climate [16,17], reproduction, morphology [18,19,20], variety [21], harvest time [22], soil pH [23], nutrient availability, and weather conditions [24].

Moreover, the post-harvest method of management [4], extraction method, and distillation time [25,26] also play an important role in essential oil composition. In the literature, there is plenty of exploration on essential oil content, composition, and various biological activities of *Lavandula* species [27,28,29], including Bulgarian cultivars [30]; however, there is a lack of comparative study on essential oil content and chemical composition of *L. angustifolia* Mill. and *L.* × *intermedia* essential oils in the mid hills of the western Himalaya under different post-harvest drying hours. The present study aims to investigate suitable post-harvest drying storage duration, which will encourage the hilly growers of the western Himalayan region and ultimately screen the quality stipulation and possible commercial value of produced essential oil.

## 2. Results

### 2.1. Plant Growth, Moisture Content, and Yield Attributes

The plant growth attributes viz., plant height (71.7 ± 3.33 cm), ear length (8.8 ± 0.20 cm), number of spikes (18.1 ± 0.41), and number of flowers per ear (47.5 ± 0.23) were significantly higher in lavandin, while the number of branches (17.1 ± 0.11) per plant was highest in lavender. Essential oil content (%) and moisture reduction (%) were significantly higher in 72 h compared to 0 h of drying in both *Lavandula* species (Figure 1). The increase in moisture reduction in 72 h was 28.59% higher, while essential oil content was 69.23% higher in lavender than in 24 and 0 h, respectively. Similarly, the increase in moisture reduction in 72 h was 20.61% and essential oil content was 61.90% higher in lavandin as compared to 24 and 0 h, respectively.

### 2.2. Essential Oil Composition

The essential oil of *Lavandula* species led to the identification of sixteen constituents, representing 92.3–95.5% of the total volatiles. The ANOVA of the data demonstrated statistically significant (*p* = 0.05) effects of *Lavandula* species and drying hours on content (%) and constituents (area%) of essential oil (Table 1). The identified constituents were similar in both the *Lavandula* species (except 1,8-cineole, isophorone, caryophyllene, delta-cadinene, and caryophyllene oxide), and differed only in the area of distribution (Table 2). The major contribution (mean) was of linalool (33.6–40.5%), followed by linalyl acetate (10.8–13.6%), lavandulyl acetate (2.8–14.5%), and linalyl propionate (5.3–14.1%) in both the *Lavandula* species. Significantly higher 3-octanone, linalool, cryptone, linalyl acetate, lavandulyl acetate, neryl acetate, and caryophyllene were recorded in lavender compared to lavandin. On the other hand, (Z)-β-ocimene, isophorone, terpinen-4-ol, and linalyl propionate were significantly higher in lavandin compared to lavender. During different drying hours, the content of 3-octanone, linalool, isophorone, and lavandulyl acetate were significantly higher in 0 drying hours, while 3-octanone remained at par with 24 drying hours. We have not recorded any regular trend in the occurrence of essential oil constituents with an increase or decrease in drying hours for *Lavandula* species.

### 2.3. Grouped Constituents Classes of Essential Oil

The grouped essential oil constituents were also significantly (*p* = 0.05) affected by *Lavandula* species and drying hours (Table 2). Significantly higher (mean) monoterpenes (50.10%) and oxygenated sesquiterpenes (1.13%) were recorded in lavandin, while sesquiterpenes were higher in lavender, but oxygenated monoterpenes were not significantly affected by *Lavandula* species. The drying hours affected the grouped constituent content distribution, but no regular trend was observed with an increase or decrease in drying hours (Table 2). Moreover, significantly higher (mean) monoterpenes (52.72%) and sesquiterpenes (3.52%) were recorded at 24 h of drying compared to 48 h and 72 h of drying, respectively. On the other hand, significantly higher oxygenated monoterpenes (47.57%) and oxygenated sesquiterpenes (0.85%) were recorded in 48 h of drying compared to 24 h of drying.

### 2.4. Principal Component Analysis

The identified essential oil constituents (>2% area) of *Lavandula* species were considered as variables and subjected to principal component (PC) analysis to evaluate the relation between *Lavandula* species and drying hours levels (Figure 2 and Figure 3). The analysis revealed 99.99% of the total variability explained by PC1 and PC2 (Figure 2a). The PC analysis of lavender separated the drying levels 0, 24, 48, and 72 h in PC1 and PC2 by presenting positive contribution and strong relationships with (Z)-β-ocimene, linalool, linalyl propionate, neryl acetate, and caryophyllene. The present PCA findings of lavender showed that the first three PCs were decidedly enlightening with 6.98, 3.02, and 0.01 as eigen values; and consequently gave about 99.99% of the overall essential oil compounds’ variance. The observed score plot of lavender possibly would be divided into four distinct clusters (Figure 2a and Table 3). The clusters were prepared from the area (%) of essential oil components through PCA software and the treatments with almost similar or related essential oil composition fall together or in the nearby location of the *X*-axis and *Y*-axis planes of the graph. The treatments that fall together were then clubbed together and given a cluster number; then, the content of essential oil components in the treatments which fall under a particular cluster was added to form the area%, as given in Table 3. Cluster I, II, III, IV corresponded to 0, 24, 48, and 72 h, respectively. Cluster I explained a higher percentage of lavandulyl acetate (30.8%), while cluster II explained a higher percentage of linalool (45.0%). Similarly, a higher percentage of linalyl acetate (34.2%) was explained by cluster III, while cluster IV observed an intermediate percentage of linalool (40.0%); this compositional variability of the treatments made them fall separate from each other and form distinctive clusters (Figure 2a and Table 3). Among all variables in lavender, 3-octanone, (Z)-β-ocimene, linalool, linalyl propionate, lavandulyl acetate, neryl acetate, and caryophyllene showed a positive association, while terpinen-4-ol, cryptone, and linalyl acetate showed negative association in PC1. Similarly, in PC2, (Z)-β-ocimene, linalool, terpinen-4-ol, cryptone, linalyl propionate, linalyl acetate, neryl acetate, and caryophyllene showed a positive association, while 3-octanone and lavandulyl acetate showed a negative association (Figure 2b). 

Similarly, the PC analysis of lavandin separated the drying levels 0, 24, 48, and 72 h in PC1 and PC2 by screening positive contribution and good relationships with linalool and isophorone. The present PCA findings of lavandin showed that the first three PCs were definitely helpful with 7.1, 1.1, and 0.01 as eigenvalues, and thus contributed to 99.36% of total essential oil compounds’ variance. Three distinct clusters for lavandin (Figure 3a and Table 3) emerged from the PC analysis of constituents. Cluster I corresponds to 0 h with the lowest linalyl propionate percentage (16.9%), while cluster II corresponds to 24 h of drying with the lowest linalool (33.0%) and highest linalyl propionate (21.9%) percentage. However, cluster III included 48 and 72 h of drying which corresponds to the highest linalool (63.5%) and highest linalyl acetate (36.2%) percentage (Figure 3a and Table 3). Among all variables in lavandin, 3-octanone, (Z)-β-ocimene, linalool, isophorone, and linalyl propionate showed a positive association, while terpinen-4-ol, cryptone, linalyl acetate, and lavandulyl acetate showed negative association in PC1. Similarly, in PC2, linalool, isophorone, and lavandulyl acetate showed a positive association, while 3-octanone, (Z)-β-ocimene, cryptone, linalyl propionate, and linalyl acetate showed a negative association (Figure 3b). On the whole, a significant difference in essential oil constituents was recorded in *Lavandula* species and varied post-harvest drying hours.

## 3. Discussion

### 3.1. Plant Growth, Moisture Reduction, and Yield Attributes

Morphological characteristics, viz., plant height, ear length, and number of spikes were slightly higher in lavandin, while a considerable increase (23.31%) in flower number per ear was observed in lavandin. The plant morphological characters of *Lavandula* cultivars in semi-arid climatic conditions of Isparta Province were earlier studied and recorded higher growth and yield attributes in cultivars of lavandin as compared to lavender [5]. Corroborating to our findings, a similar growth attribute was reported in lavender when grown in subtropical conditions in Italy [31].

The highest moisture reduction was recorded in 72 h of drying in both *Lavandula* species because the rate of drying was much faster from 0 to 24 h, but the rate gradually decreased with later drying hours from 24 to 72 h. Similar findings in moisture reduction have been reported in *Pistacia lentiscus* L. leaves with an increase in drying time [32]. The highest essential oil content was obtained in 72 h of drying and the findings were in good agreement with findings in *Eugenia uniflora* L. leaves where higher essential oil content was recorded in later drying of 48 h [33]. The volatile oil content has been reported to be affected by the drying time, drying method, temperature, and plant species because the moisture level of samples dried for different times can be highly variable and affected essential oil content [34]. For instance, the highest essential oil content was recorded in 72 h of drying when moisture reduction was maximum (Figure 1) in samples compared to other drying hours. In addition, some important physiological activity (enzymatic reactions) that continue after crop harvest was responsible for the biosynthetic activity of terpenes and resulted in an increase in essential oils content [35] reported within the limits as in earlier findings in *Lavandula* species [36,37]. A prolonged drying time (up to 8 days) decreased the essential oil content of kaffir lime (*Citrus hystrix* DC), but shorter drying from zero to four days increased the essential oil content [38], which is similar to present findings up to 72 h of drying. The shorter shade drying period (first seven days) increased essential oil content, but decreased with increased drying duration [32].

### 3.2. Essential Oil Composition

The identified sixteen essential oil constituents were significantly influenced by *Lavandula* species and post-harvest drying hours (Table 1). The major contribution in the essential oil of both *Lavandula* species was linalool and linalyl acetate, and overall content ranged from 32.9% to 45.0% and 0.6% to 34.2%, respectively. The earlier findings have also reported linalool and linalyl acetate as 10.0–57.5% and 4–55%, respectively, in the essential oil of *L. angustifolia* [36], while linalool, linalyl acetate, and borneol were shown to be major essential oil constituents in lavandin [5].

The presence of linalool and linalyl acetate is the criterion for the quality of lavender essential oil and is highly demanded in the international markets because of their sedative and narcotic effects [39]. The content of camphor in the present finding was even less than 0.1% among the drying hours and not shown in Table 1,which corroborated with the International Organization for Standardization (ISO 3515:2002) for camphor content in lavender, i.e., lower than 0.5%. In the present study, the lavender and lavandin recorded higher linalool and linalyl acetate at 48 h of drying and were according to essential oil standards (ISO 3515:2002). Moreover, long-term storage of dried flowers reduced the total constituents’ content viz., linalool, linalyl acetate, lavandulyl acetate, and terpinen-4-ol between the lavender varieties [37]. The constituents of *L. angustifolia* and *L.* × *intermedia* obtained from essential oil in the present finding ranged from 3-octanone: 1.9–2.5; limonene: 0.6–1.0%; 1,8-cineole: 0.0–2.7; (Z)-β-ocimene: 1.4–15.6%; linalool: 30.7–45.0%; terpinen-4-ol: 0.5–7.3%; linalyl acetate:3.0–34.2%; and lavandulyl acetate: 1.6–30.8%, and qualifies the acceptable ranges of ISO Standard 3515:2002 [40], except for a few essential oil constituents. The variability in essential oil depends on several factors, viz., environment, climate, geographical location, season, genetic makeup, chemotype, plants’ nutritional status, the extraction process, and the plant part used [41].

There was a considerable decrease in the content of lavandulyl acetate with an increase in drying hours from 0 to 48 h, and again showing an increase at 72 h of drying in both *Lavandula* species. The decreased lavandulyl acetate can be explained as changes that have resulted because of the conversion of lavandulyl and linalyl esters to alcohols and molecular reshuffling [37]. The statistical analysis of the drying duration on the essential oil composition does not confirm the increasing or decreasing trends for all essential oil constituents. However, it can be assumed that if a longer drying period was evaluated, differences in the composition of essential oil would be confirmed statistically. The constituents of lavandula essential oil are not only linalool, linalyl acetate, and lavandulyl acetate, but a complex of up to 60 monoterpenes and sesquiterpenes [17]. The interactions of all of these substances with their moisture content at the time of hydrodistillation determine the resulting chemical composition of the essential oil.

Moreover, in the process of hydrodistillation, the degradation/hydrolysis of thermo-labile compounds/water-sensitive compounds has been observed, and the esters are generally hydrolyzed by the release of H^+^ ions from water to analogous acids and alcohols. The pH of water is often reduced during hydrodistillation and essential oil components like esters are sensitive to hydrolysis, while others like acyclic monoterpene hydrocarbons and aldehydes are susceptible to polymerization because of low pH [42]. As the terpenoids are volatile and thermolabile, they may be easily oxidized or hydrolyzed depending on their respective structure. The chemical composition of essential oils is additionally dependent on the conditions of storage of the plant material, processing conditions upon distillation, and after extraction storage conditions [43]. This degradation and rearrangement resulted in the increase and decrease of essential oil constituents of *Lavandula* species in the present study. The conversion of a few essential oil constituents such as geranyl acetate to geraniol, neral, and geranial was also observed during the storage of ginger (*Zingiber officinale* R.) [44]. The data obtained in the present study was valuable to respond to the issue of whether there is any loss in essential oil constituents during post-harvest drying of *Lavandula* species. The interactions among all the constituents determine the resultant chemical properties of the essential oils, and thus provide evidence of whether there was a loss of essential oil composition at the initial and at the final stage of drying.

The *Lavandula* species studied in the present study recorded a higher percentage (mean) of monoterpenes and oxygenated sesquiterpenes (1.13%) in lavandin (50.10%), and higher oxygenated monoterpenes (42.64%) and sesquiterpenes (3.78%) in lavender. This diversity in mono- and sesquiterpene can be associated with the catalytic activity of the enzymes of the terpene synthase (TPS) family. The complex spatial and temporal arrangement of TPS members during plant development and biotic/abiotic stresses may be responsible for the expression profile of terpenes [45]. The activities of TPS were shown to be regulated at the transcription level [45], and in peppermint, the transcription of TPS and numerous terpene biosynthetic genes get modified during the development stage [46] and regulate terpene biosynthesis [47]. Similar to our findings, hydrodistillation usually produced essential oil with chemical composition chiefly described by the presence of oxygenated monoterpenes viz., linalool, linalyl acetate, 1,8-cineole, and monoterpenoids, viz., lavandulyl acetate. Monoterpenes viz., α-β-pinene and β-ocimene; sesquiterpenes, viz., caryophyllene; and oxygenated sesquiterpenes viz., caryophyllene oxide were represented as minor compounds [48], while some described molecules of several chemical derivatives (phenolic acids and flavonoid) [49,50]. Monoterpenes and sesquiterpenes are the most recurrently studied hydrocarbons in lavender essential oil due to their abundance and importance in the fragrance and flavor industry [51]. Corroborating to our findings, lavender and lavandin produced the majority of oxygenated monoterpenes followed by monoterpene hydrocarbons with linalool and linalyl acetate as major constituents [52,53].

## 4. Materials and Methods

### 4.1. Experimental Site

An experiment was executed in 2020 in the CSIR-IHBT (Council of Scientific and Industrial Research, Institute of Himalayan Bioresource Technology), Palampur, Himachal Pradesh (HP), India; situated at an altitude of 1390 m above mean sea level (amsl) (32°11′39″ N latitude and 76°56′51″ E longitude) in a subtropical climate. Weather parameters viz., minimum and maximum temperature (°C), and relative humidity (RH%) during the third year of crop growth are depicted in Figure 4. The minimum temperature ranged from 9 to 26 °C, while the maximum temperature ranged from 19 to 37 °C during the crop harvesting year. Mean relative humidity (%) ranged from 63% to 98% during crop growth.

### 4.2. Treatment Details

The experiment consists of eight treatments, two *Lavandula* species viz., common lavender (*L. angustifolia* Mill.) and lavandin (*Lavandula* × *intermedia* Emeric ex Loisel), and four drying hours, i.e., 0 h, 24 h, 48 h, and 72 h, with three replications. *Lavandula* species were raised in polythene tunnel conditions, planted in October 2016, and harvested in May 2020. Both the *Lavandula* species were planted in a polythene tunnel because the experimental location is a high rainfall area, and thus, the species were protected from water logging conditions in the polythene tunnel. The polythene tunnel was 23.30 × 3.90 m for lavender and 14.50 m × 1.80 m for lavandin. The soil was a mixture of sand, soil, and decomposed farmyard manure (FYM at a rate of 20 kg m^−2^).

### 4.3. Experimental and Plant Details

The soil samples collected at 0–20 cm depth from the experimental tunnels were pooled, dried, sieved (2 mm sieve), and stored at room temperature until soil physicochemical analysis. The soil of the experimental polythene tunnel area was clay loam in texture, acidic pH (5.3), medium in available N (184.3 kg ha^−1^), and available P_2_O_5_ (16.4 kg ha^−1^), while high in available K_2_O (478.0 kg ha^−1^). The plants were transplanted in October 2016 and harvested at the full flowering stage in May 2020. The fresh plant material of both *Lavandula* species was harvested and then shade dried under conventional open-air conditions for 0, 24, 48, and 72 h after harvesting, and then essential oil was extracted. The minimum and maximum daily temperature during five drying hours were between 14–20 °C and 29–33 °C, respectively, while humidity ranged from 49–73% during crop drying conditions. The harvested biomass (weight: 1 kg) of both the *Lavandula* species was spread over a clean and dry surface area (1 m^2^) for each drying hour treatment.

### 4.4. Plant Growth Investigation

Plant height was measured in centimeters (cm) by randomly selecting the plants (five plants per species) from the base of the ground to the tip of the main stem by using measuring tape; then, the average height of the plant was determined. The branch number and number of spikes per plant arising from the main stem were determined by counting the stems manually, and the average value was determined. Ear length per spike and number of flowers per ear were also determined through scale and manual counting, respectively, at harvest.

### 4.5. Identification and Determination of Essential Oil Components

From each treatment, fresh samples were harvested, and then brought to the laboratory and hydrodistilled after 0, 24, 48, and 72 h of drying in triplicates for 4 h in the Clevenger-type apparatus [54]. The essential oil content was calculated as volume/weight (%), i.e., volume (mL) of essential oil obtained per weight (g) of flowers and then dried with anhydrous sodium sulfate (Na_2_SO_4_) to remove any water droplet. The essential oil was collected in a glass container and stored at 4 °C before analysis. GC and GC/MS analysis were performed in triplicates through a flame ionization detector (FID) on Shimadzu GC 2010 gas chromatography and QP2010 (Shimadzu Corp., Tokyo, Japan) fitted with an AOC 5000 auto-injector. The auto-injector comprises a 30 m long ZB-5 MS capillary column with a 0.25 mm i.d. and a 0.25-μm thick film (SGE International, Ringwood, Australia). Essential oil (10 μL) was dissolved in 2 mL of dichloromethane and auto-injected with 2 μL volume in split mode. Carrier gas (nitrogen) was used with a 1.05 mL min^−1^ flow rate; an oven temperature of 70 °C for 3 min was used, which was afterward raised to 220 °C for 5 min at the rate of 4 °C min^−1^, and the injector and detector temperature was programmed at 220 °C and 250 °C, respectively. The temperature programming, injection volume, and carrier gas conditions utilized for performing GC and GC/MS were according to the mentioned procedure in *Matricaria chamomilla* L. [55]. To identify the compounds from GC peak areas, a series of hydrocarbons was utilized for retention index (RI) estimation. The essential oil components were identified after matching the experimental RIs with the RIs in the literature [56]. Furthermore, the components were identified by matching the minimum mass spectral fragmentation pattern of the components with the National Institute of Standards and Technology (NIST) library [57].

### 4.6. Statistical Analysis

After chemical evaluation, data were analyzed using the analysis of variance (ANOVA) technique for factorial Complete Random Design (CRD). Fisher’s least significant difference (LSD) test was performed for the least significant testing and regarded statistically significant at *p* = 0.05. The essential oil constituents were subjected to multivariate principal component analysis (PCA software PAST 3) to assess the expression of treatments on essential oil constituents.

## 5. Conclusions

The current experiment assessed the changes in the essential oil content and composition after 0, 24, 48, and 72 h of post-harvest shade drying in *Lavandula* species under mid hill conditions of the western Himalaya. Present findings confirmed inter-specific variability in *Lavandula*; specific drying duration ought to be taken into consideration to optimize essential oil production and industrial utilization. *Lavandula* species recorded significantly higher essential oil content at 72 h of drying, and acceptable quality (ISO 3515:2002 and ISO 8902:2009) of essential oil in terms of linalool and linalyl acetate content. During drying duration, lavandin recorded a decreasing trend in the linalool content and an increasing trend of linalyl acetate, while lavender recorded an irregular trend in linalool and linalyl acetate with drying duration. The differences in essential oil content and chemical constituents were statistically confirmed, although the confirmation was for a short-term drying duration of up to 72 h of monitoring. Nevertheless, investigations should be confirmed by drying the *Lavandula* species for a longer duration as the content of essential oil increased up to 72 h of drying and can be carried out for a longer duration to evaluate essential oil content and composition. However, it is necessary to conduct further research aimed at developing improved agronomic practices and post-harvest management for improved yield and desired quality of essential oil.

## Figures and Tables

**Figure 1 molecules-27-03391-f001:**
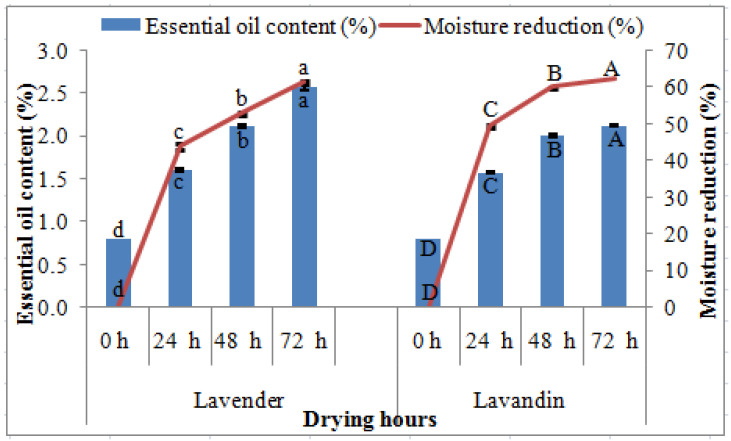
Effect of *Lavandula* species and drying hours on moisture reduction (%) and essential oil content (%). Note: Dissimilar letters represent significant differences in moisture reduction (%) essential oil content (%) at *p* = 0.05.

**Figure 2 molecules-27-03391-f002:**
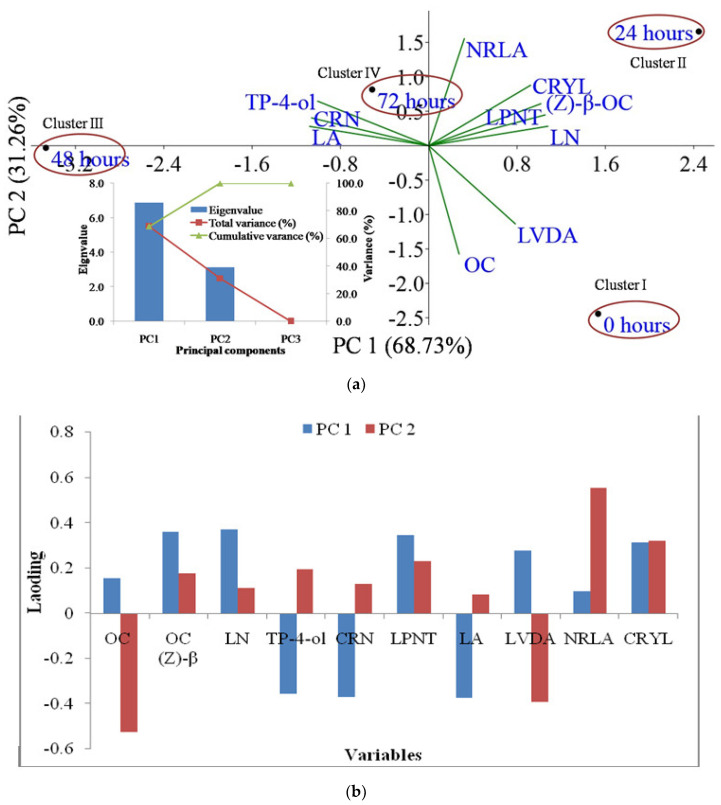
Principal component 1 and principal component 2 jointly explained 99.99% of the total variation for lavender and storage hours; eigenvalues are presented in the scree plot in the bottom left corner of the figure (**a**), while a loading plot of principal component analysis of lavender and storage hours is presented in figure (**b**). Note: OC: 3-octanone; OC (Z)-β: (Z)-β-ocimene; LN: linalool; TP-4-ol: terpinen-4-ol; CRN: cryptone; LPNT: linalyl propionate; LA: linalyl acetate; LVDA: lavandulyl acetate; NRLA: neryl acetate; CRYL: caryophyllene.

**Figure 3 molecules-27-03391-f003:**
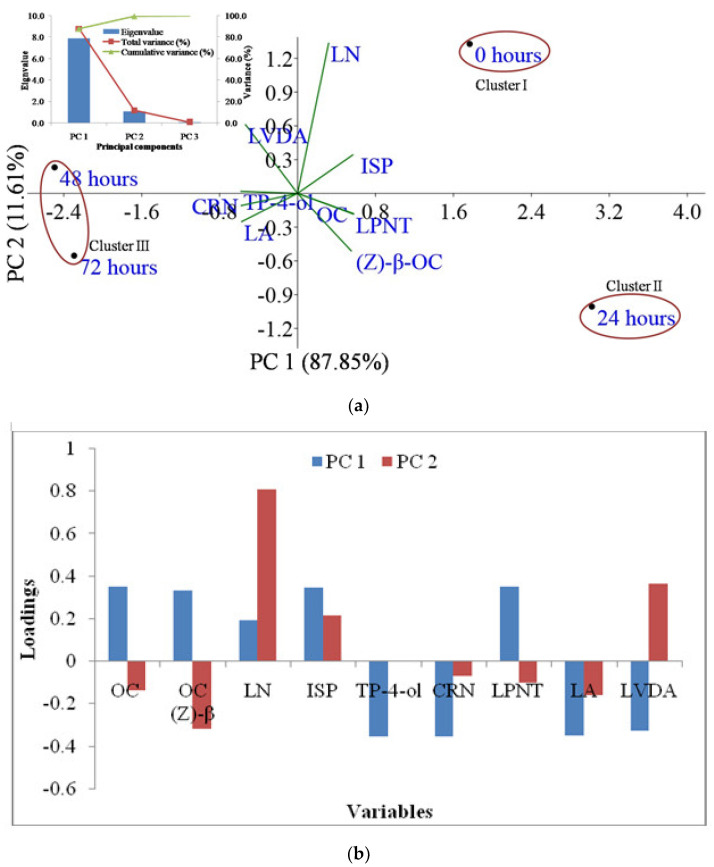
Principal component 1 and principal component 2 jointly explained 99.36% of the total variation for lavandin and storage hours; eigenvalues are presented in the scree plot in the top left corner of the figure (**a**), while a loading plot of principal component analysis of lavandin and storage hours is presented in figure (**b**). Note: OC: 3-octanone; OC (Z)-β: (Z)-β-ocimene; LN: linalool; ISP: isophorone; TP-4-ol: terpinen-4-ol; CRN: cryptone; LPNT: linalyl propionate; LA: linalyl acetate; LVDA: lavandulyl acetate.

**Figure 4 molecules-27-03391-f004:**
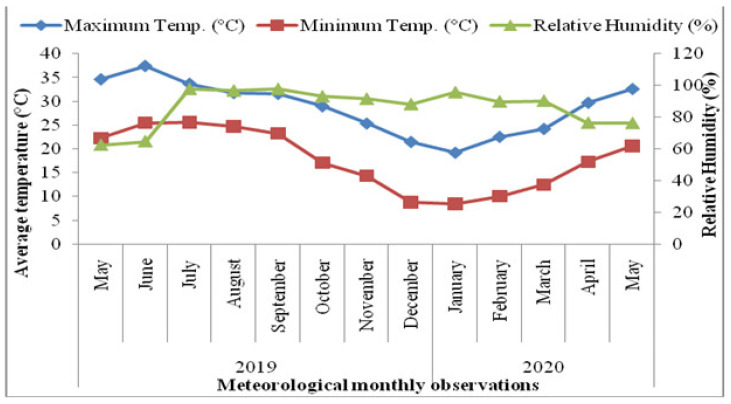
Average monthly meteorological observations from 2019–2020 under polyhouse conditions of Palampur, India.

**Table 1 molecules-27-03391-t001:** Effect of lavender species and storage hours on essential oil composition (area%).

Treatment	Constituents															
	3-Octanone	α-Phellandrene	Limonene	1,8-Cineole	(Z)-β-Ocimene	Linalool	Isophorone	Terpinen-4-ol	Cryptone	Linalyl Propionate	Linalyl Acetate	Lavandulyl Acetate	Neryl Acetate	Caryophyllene	Delta-Cadinene	Caryophyllene Oxide
RI (Expt.)	982	1004	1025	1027	1030	1097	1125	1173	1187	1223	1250	1279	1350	1409	1547	1578
RI (Lit.)	979	1002	1024	1026	1032	1095	1118	1174	1183	1214	1254	1288	1359	1408	1522	1582
*L. angustifolia*																
Drying hours																
0 h	2.5	0.3	0.6	0.0	3.3	41.9	0.0	0.5	2.6	5.4	0.6	30.8	3.0	2.9	0.6	0.0
24 h	2.4	0.0	1.0	0.0	4.6	45.0	0.0	0.9	3.0	9.9	1.5	14.7	4.5	6.3	0.8	0.0
48 h	2.4	0.2	0.6	2.7	1.4	35.1	0.0	2.2	6.3	0.7	34.2	3.6	3.5	1.0	0.0	0.0
72 h	2.4	0.0	0.8	1.4	3.0	40.0	0.0	1.5	4.7	5.3	17.9	9.1	4.0	3.6	0.0	0.0
Mean	2.4	0.1	0.8	1.0	3.1	40.5	0.0	1.3	4.2	5.3	13.6	14.5	3.8	3.4	0.3	0.0
*L.* × *intermedia*																
Drying hours																
0 h	2.3	0.3	1.0	0.0	13.5	38.0	10.7	4.7	0.6	16.9	3.7	2.8	0.6	0.0	0.0	0.9
24 h	2.5	0.5	1.0	0.0	15.6	33.0	10.2	3.8	0.0	21.9	3.0	1.6	0.9	0.0	0.0	0.3
48 h	1.9	0.0	0.8	0.0	11.5	32.9	0.0	7.3	4.6	9.0	17.9	3.5	2.4	0.0	0.0	1.7
72 h	2.0	0.0	0.9	0.0	12.2	30.7	0.0	7.3	4.2	8.6	18.4	3.4	1.5	0.0	0.0	1.7
Mean	2.2	0.2	0.9	0.0	13.2	33.6	5.2	5.8	2.3	14.1	10.8	2.8	1.3	0.0	0.0	1.1
Mean (DH)																
0 h	2.4	0.3	0.8	0.0	8.4	39.9	5.4	2.6	1.6	11.2	2.2	16.8	1.8	1.4	0.3	0.4
24 h	2.4	0.2	1.0	0.0	10.1	39.0	5.1	2.4	1.5	15.9	2.3	8.1	2.7	3.1	0.4	0.2
48 h	2.1	0.1	0.7	1.4	6.4	34.0	0.0	4.7	5.5	4.9	26.1	3.5	2.9	0.5	0.0	0.8
72 h	2.2	0.0	0.8	0.7	7.6	35.4	0.0	4.4	4.4	6.9	18.1	6.3	2.8	1.8	0.0	0.8
LSD (*p* = 0.05)																
LS	0.1	0.0	0.0	0.0	0.2	0.2	0.0	0.1	0.1	0.2	0.5	0.3	0.1	0.3	0.3	0.2
DH	0.1	0.0	0.0	0.0	0.2	0.3	0.0	0.1	0.2	0.2	0.7	0.4	0.2	0.4	NS	0.2
LS × DH	0.1	0.0	0.0	0.0	0.4	0.4	0.0	0.2	0.3	0.3	0.9	0.6	0.3	0.6	NS	0.3

LSD: least significant difference (*p* = 0.05); LS: *Lavandula* species; DH: drying hours; Exp. RI (GC): experimental retention indices (gas chromatography); Exp. RI (GC/MS): experimental retention indices (gas chromatography/mass spectrometry).

**Table 2 molecules-27-03391-t002:** Effect of lavender species and storage hours on classes of grouped constituents (area %).

Treatment	Monoterpenes	Oxygenated Monoterpenes	Sesquiterpenes	Oxygenated Sesquiterpenes
*L. angustifolia*				
Drying hours				
0 h	48.62	42.89	3.47	0.00
24 h	52.97	34.59	7.05	0.00
48 h	42.47	50.52	0.96	0.00
72 h	47.64	42.56	3.63	0.00
Mean	47.93	42.64	3.78	0.00
*L.* × *intermedia*				
Drying hours				
0 h	55.17	40.02	0.00	0.85
24 h	52.46	41.40	0.00	0.31
48 h	47.06	44.61	0.00	1.69
72 h	45.69	43.33	0.00	1.69
Mean	50.10	42.34	0.00	1.13
Mean (DH)				
0 h	51.90	41.45	1.74	0.43
24 h	52.72	38.00	3.52	0.15
48 h	44.77	47.57	0.48	0.85
72 h	46.66	42.94	1.81	0.84
LSD (*p* = 0.05)				
LS	0.31	NS	0.39	0.16
DH	0.44	0.80	0.55	0.23
LS × DH	0.62	1.13	0.78	0.33

LSD: least significant difference (*p* = 0.05); LS: *Lavandula* species; DH: drying hours. Monoterpenes: 3-octanone, α-phellandrene, limonene, 1,8-cineole, (Z)-β-ocimene, and linalool. Oxygenated monoterpenes: isophorone, terpinen-4-ol, cryptone, linalyl propionate, linalyl acetate, lavandulyl acetate, and neryl acetate. Sesquiterpenes: caryophyllene and delta-cadinene. Oxygenated Sesquiterpenes: caryophyllene oxide.

**Table 3 molecules-27-03391-t003:** Clusters variability of major essential oil constituents (>2%) in essential oil of lavender and lavandin species with storage hours.

Essential Oil Constituents	*Lavandula* Species						
	*L. angustifolia*				*L.* × *intermedia*		
	Cluster I	Cluster II	Cluster III	Cluster IV	Cluster I	Cluster II	Cluster III
3-octanone	2.5	2.4	2.4	2.4	2.3	2.5	3.9
(Z)-β-ocimene	3.3	4.6	1.4	3.0	13.5	15.6	23.6
Linalool	41.9	45.0	35.1	40.0	38.0	33.0	63.5
Isophorone	0.0	0.0	0.0	0.0	10.7	10.2	0.0
Terpinen-4-ol	0.5	0.9	2.2	1.5	4.7	3.8	14.6
Cryptone	2.6	3.0	6.3	4.7	0.6	0.0	8.8
Linalyl propionate	5.4	9.9	0.7	5.3	16.9	21.9	17.6
Linalyl acetate	0.6	1.5	34.2	17.9	3.7	3.0	36.2
Lavandulyl acetate	30.8	14.7	3.6	9.1	2.8	1.6	6.8
Neryl acetate	3.0	4.5	3.5	4.0	0.0	0.0	0.0
Caryophyllene	2.9	6.3	1.0	3.6	0.0	0.0	0.0

## Data Availability

The data supporting the reported results is available with the authors.

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
