# Peer review of "Essential Oil Content and Compositional Variability of Lavandula Species Cultivated in the Mid Hill Conditions of the Western Himalaya"

_molecules, 2022, doi:10.3390/molecules27113391_

Round 1

Reviewer 1 Report

The charts are not very legible and not very aesthetic.

I understand that the tables only show the percentages of the individual compounds? It is not possible to give specific concentrations?

Page 8, lines 221-229 - it could use an explanation as to where this is coming from, no discussion.

In general, in the Discussion chapter, I would like to ask you to try to explain the individual results, what they may result from - this is how I understand the discussion of results as an attempt to understand a given phenomenon or process.

It seems to me that the the Results chapter could be shortened. You describe the results in great detail, but remember that there are also charts, tables - so there is no need to describe everything. Please pay attention to the most important points that you later discuss.

(FYM at a rate of 20 357 kg m2) - kg/m2

Reviewer 2 Report

Reference: molecules-1691171

The manuscript "Essential oil content and compositional variability of Lavandula species cultivated in the mid hill conditions of the western Himalaya" by Rathore and Kumar presents the evaluation of the quantity of some components according to the drying period (0-72 h).
The work is not very innovative but it has some data that can be of interest, since the species atudies has market value. However, it is logical that the % yield of the plant components tends to increase when the
water content decreases. Although the manuscript presents and discusses some other data, this obvious part takes too much space in my opinion. The manuscript
would be more interesting if the authors concentrate their attention in the other findings. I would strongly advise the authors to shorten the mansucript, to
highlight data different that yield increase over the drying period. Language is a problem in the manuscript, either because of some grammar erros, or due prolix discussions/results presentation.
Again, I suggest full revision. I call attention of some specific details above.

I suggest to add a reference abot MAPs to substitute reference [1], since reference 1 talks about a specific example and not about a general condition.
Line 38: Substitute: by an appropriate verb.
Line 41: "... by France; the top..."
Line 46. Reference [4] is an specific example, while the authors are referrinf to a broad situation. Please, rephrase.
Line 53. The yield of essential oil is
Line 55 [2,3,6], instead of [3,2,6]. Regarding the monetary value of the kilogram of the oils, please, add recent values, since there was a huge variation in the market due covid-19. Otherwise, 
rephrase to inform that you mention data from 2018.
Line 57: due to its relatively higher camphor content 
Line 75. Please, rephrase: western the western

2. Results
Lines 79-81. Please, remove the template instructions
Figure 1. Please, use the correct international abbreviation for hours (h); Since data for lavander and lavandin are independent, the red courve should have a break between 72 h 
for lavander and 0 h for lavandin
Lines 87-89: this part is very confusing: "The increase in moisture reduction... was... higher". Please, rephrase here and in the following sentence. Pay attention to because it is also not clear to which parameter the % values refer.
You mention two parameters (moisture reduction and essential oil content) at 72 h, compared to two periods of time (24 and 0 h), but give only two % values.
Lines 104 and followings. Please, correct the nomenclature. Use 1,8-cineole instead of  cineole <1,8->; the same for the compiunds cited in lines 109, 111, 113, 115 and others.
According to the international rules, it is not correct to write part of the nomenclature using <>.
Line 123 Table 1. Please, remove dot after the word Table
Tale 1. (litt) refers to the literature? If it does, please, use (lit) and add the reference. In Table 1, page 4, a heading to explain that you start presenting % must be added; otherwise, it is confusing.
Lines 124 and 125. Please, remove the data of abbreviatures that were not used in the table. Also, if you want to show the % diference after 0, 24, 48 and 72 h of drying, why did you add the mean? And also, I could not understand what
are you refering in the lines corresponding to "Mean (DH)"

Table 2. Please, indicate in the bottom of the table, which are the monoterpenes, oxygenated monoterpenes, etc. Here, again, what is the meaning of "Mean (DH)"
Figure 2. The PCA chart must be edited to become readble. 
Lines 175-177: Please, transfer the "Note" to the bottom of the figure 
Line 281-288. Language improvement is necessary; lines 282-283. 7.11 is not high; but it can be higher than... 
Line 292. Is lavandulyl acetate the most volatile constituent analysed? Is it so different in terms of volatility to explain the diference? As far as I know, several of the other components are also volatile.
It is unlike that lavandulyl acetate is hydrolized, while linalyl acetate is not; insteady, the concentration of linalyl acetate 
increases. I suggest that authors think deeper about this paraghaph and remove especulative data. The increase of the concentration of some compounds is probably related to 
the decrease in the water content.
Line 295. the, instead of there
Line 297. The hydrolisis may be related to the heating also, since the acidity seems to be very low.

Please, add an space between the value and the oC unity. Check all over the manuscript, for example, in lines 339 and 369.
Once referring to Lavandula as a genus, please, add italics. 

Reviewer 3 Report

Dears authors

The increase in the use of Lavandula essential oil in industries has led to an impressive increase in the demand for quality essential oils. The dilemma is to stumble upon a post-harvest drying of the Lavandula species to obtain a better quantity and quality of essential oil. The study was conducted in the conditions of the western Himalayas to evaluate the essential oil content and composition of two species of La-vandula, namely, lavender (Lavandula angustifolia Mill.) And Lavender (Lavandula × intermedia Emeric ex Loisel) , at four different drying times (0 hours, 24 hours, 48 ​​hours and 72 hours after the harvest). The highest growth attributes i.e. plant height (71.7 cm), ear length (8.8 cm), number of spikes (18.1) and number of flowers per spike (47, 5) were higher in lavender while the number of branches (17.1) was higher in lavender. Essential oil content (%) and moisture reduction (%) were significantly higher at 72 hours than at 0 hours. The main components of Lavender and Lavender essential oil were Linalool (33.6 - 40.5%), Linalyl Acetate (10.8 - 13.6%), Lavendulyl Acetate (2.8 - 14.5%) and linalyl propionate (5.3 - 14.1%) in both lavender species. There was a decreasing trend in linalool and an increasing trend in linalyl acetate content with an increase in drying duration up to 72 hours while a regular trend in linalool and linalyl acetate content was not observed in lavender.
As a summary assessment, the paper is well articulated
Learn more:
- Insert a map with the collection points of the various lavender specialties considered.
- In table 3 and the clusters, deepen them with greater explanation of the components and also insert them in the text which analysis technique was used?
- Deepen in the discussion the use given the lack of antimicrobials of the use of essential oil of lavender as an antimicrobial and clinical application:

PMID: 31628828, PMID: 34579362 , PMID: 35268571 

Round 2

Reviewer 2 Report

Dear Authors,

I am satisfied with the modifications made in the manuscript. The current version is clear, speculative details were removed, and the other corrections were made.

I still think that Figures 2 and 3 could be improved, but all other aspects are ok. 

Sincerely,